# Pandemic of Childhood Myopia. Could New Indoor LED Lighting Be Part of the Solution?

**David Baeza Moyano** [1,*] and **Roberto Alonso González-Lezcano** [2]

1   Department of Chemistry and Biochemistry, Campus Montepríncipe, Universidad San Pablo CEU, 28668 Alcorcón, Madrid, Spain
2   Arquitecture and Design Deperment, Escuela Politécnica Superior, Campus Montpríncipe, Universidad San Pablo CEU, 28668 Alcorcón, Madrid, Spain; rgonzalezcano@ceu.es
*   Correspondence: baezams@ceu.es

**Abstract:** The existence of a growing myopia pandemic is an unquestionable fact for health authorities around the world. Different possible causes have been put forward over the years, such as a possible genetic origin, the current excess of children's close-up work compared to previous stages in history, insufficient natural light, or a multifactorial cause. Scientists are looking for different possible solutions to alleviate it, such as a reduction of time or a greater distance for children's work, the use of drugs, optometric correction methods, surgical procedures, and spending more time outdoors. There is a growing number of articles suggesting insufficient natural light as a possible cause of the increasing levels of childhood myopia around the globe. Technological progress in the world of lighting is making it possible to have more monochromatic LED emission peaks, and because of this, it is possible to create spectral distributions of visible light that increasingly resemble natural light in the visible range. The possibility of creating indoor luminaires that emit throughout the visible spectrum from purple to infrared can now be a reality that could offer a new avenue of research to fight this pandemic.

**Keywords:** daylighting; circadian lighting; indoor lighting; dopamine; myopia

## 1. Introduction

The development of the components of the visual system during infancy and early childhood appears to be biphasic, with emmetropization occurring within the first 2 years of infancy during a rapid exponential phase [1].

Myopia in humans consists of decompensation of the refractive power of the cornea and lens compared to the axial length, such that images lie in front of the retina [2,3]. The components of the ocular system (axial length and anterior and posterior chamber depths) during emmetropization of children aged 3 to 9 months are currently longer than those measured in previous studies [4]. Some authors define myopia as a multifactorial disorder controlled by genetic interactions and environmental risk factors [5].

Myopia is a growing major public health problem and is the world's largest refractive problem [6–9]. The richest countries in the Asia–Pacific region have the highest prevalence in the world. The are multiple studies on the incidence of myopia worldwide according to ethnic and geographic parameters, with myopia prevalence at 80–90% in young adults and high myopia rates of 10–20% [10] in the urban populations in East Asia, especially in China and South Korea [11,12], Taiwan [13], and Singapore [14], while more than one-third of the people from the United States suffer from it [15–17]. Rates differ among people living in urban or rural communities, with similarities in other respects [6].

How myopia and high myopia are defined depends on the prevalence studies selected. Some projections state that 50% of the population will have myopia and 10% will have high myopia within thirty years [2,6,17].

It has been associated with an increased risk of retinal detachment, macular degeneration, early-onset glaucoma, and cataracts [6,8,12,18] and is the leading cause of blindness worldwide [6,19,20].

Data vary depending on the source; however, this social health problem is of gigantic proportions. In Europe, the above-mentioned levels have not yet been reached; however, the percentage of children and adolescents with myopia continues to increase [21,22].

The scientific community, to date, has not been able to obtain conclusive results regarding its cause, making it difficult to find a solution.

Because of this, there has been an abundance of studies from around the world. A representative example is the Cochrane Database of Systematic Reviews, which includes 41 studies (6772 participants) evaluating the effects acting on myopia progression in children with optometric corrections, and pharmaceutical procedures, such as muscarinic receptor antagonists, cycloplegic eye drops, and intraocular pressure-lowering medications [2].

The aim of this paper is to analyze the different possible causes of this pandemic, which scientists have highlighted in recent years, and to raise the possibility that something can be done to attenuate its progression based on new discoveries about the influence of light absorbed through our eyes on multiple biochemical mechanisms in our organisms, including the retina.

## 2. Materials and Methods

A bibliographic search was conducted in the biomedical databases of PubMed and ScienceDirect. Variations of the words "myopia", "nearsightedness", "short-sight", "progression", "refractive error", "Dopamine", "prevalence" were used as search criteria. These were used in combination with the inclusion criteria "risk factors", "outside", "outdoor", "atropine", "schoolchildren", "eye growth, "circadian rhythms", "indoor LED lighting".

Figure 1 below shows the flow diagram of the process carried out for the development of this article.

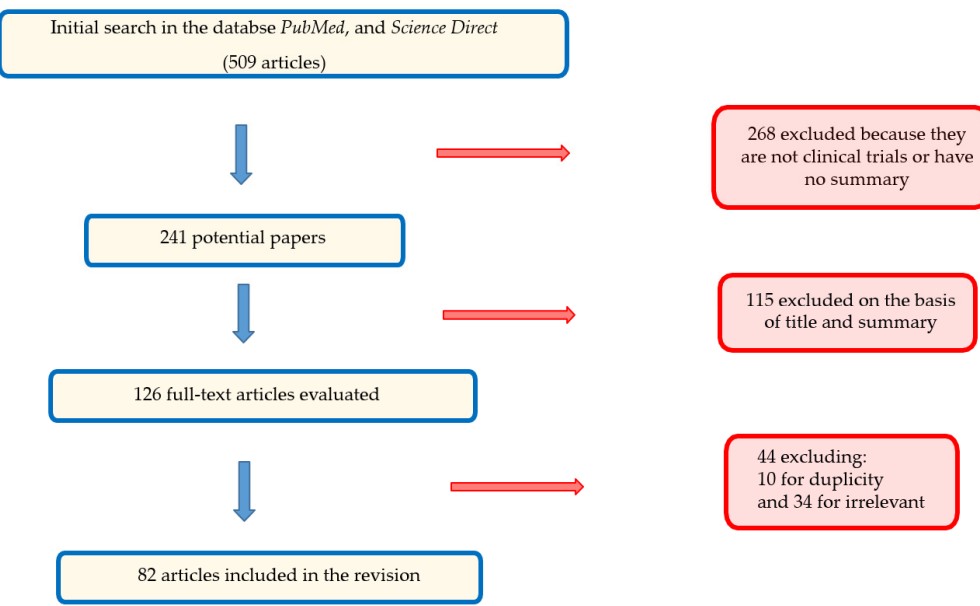

**Figure 1.** PRISMA flow chart for literature search and selection of articles included in the literature review Pandemic of Childhood Myopia. Could new indoor LED lighting be part of the solution?

The following inclusion criteria were used for the selection of selected articles: Scientific articles from the last two years, publications dealing with indoor lighting and new luminaires, indoor lighting standards, and magazines and articles written in English.

With regard to the exclusion criteria taken into account for the preparation of the work, the following were excluded publications that do not have the keyword "myopia" or "LED".

## 3. Results

Heritability is one of the factors most linked with young myopia together with increased near-distance work, increased school performance, and decreased sporting activity; however, there is no evidence that children inherit a myopathic environment or a susceptibility to the effects of near-distance work performed by their parents [23]. Children with myopic parents who do less sport and spend less time receiving natural light have a greater risk of becoming myopic than children who are in the same situation but for whom one or neither of their parents who are myopic [24]. The existence of this pandemic may be due to the pressure of increased school performance and limited outdoor time rather than a heightened sensitivity to these factors [10]. Increasing prevalence in many parts of the world makes it unlikely that its cause is related to a genetic factor [25].

Multiple factors have been raised that could influence the development of myopia in children, such as access to education, access to outdoor activities, and lack of exposure to natural light [6,26,27].

### 3.1. Myopia and Overwork at Close Distances

Increased working hours at close distances could increase the prevalence of myopia. Near-distance work has been defined as the group of activities that are carried out in proximal and to intermediate distances such as watching TV [14,27].

A higher incidence has been found in Chinese children living in Singapore compared to others living in Xiamen, because of positive relationships between the number of hours of reading, hours measured in tasks at close distances, use of electronic devices, and having high myopia; however, it is not certain that other factors do not play a role [14].

The dominant theory supported by numerous epidemiological studies about the origin of myopia is associated with near-distance work during eye development; however, there are doubts about this, as there are experimental animal studies in which no involvement of accommodation in the development of myopia is found. The site of action of atropine, which blocks myopia progression in humans, does not act on accommodation [3,10].

There are studies suggesting that the relationship between mass schooling and the onset and progression of myopia lies in the short focal distance required for reading and writing [22,28], whereby increased time spent in education may inadvertently increase the chances of suffering from myopia [29].

Others are of the opinion that, generally, no association has been found between working at close distances and myopia, except for the subgroup of people with high levels of nearby work and moderate levels of outdoor activity. A weak protective effect of outdoor activity on myopia was observed in children in rural China [30]. A high percentage of refractive problems was found in Jewish children attending orthodox schools. It has not been seen in other population subgroups, and it is very similar to that seen in East and Southeast Asia. Similar refractive distributions have also been reported for recent cohorts of young adults in Singapore [31] and South Korea [10,32].

### 3.2. Medication to Treat the Progression of Myopia

The use of atropine [16] and anticholinergic blockers has been shown to be suitable in the control of the progression myopia; however, there is considerable debate about their effectiveness and the possible long-term side effects [3].

Many researchers thought that excess accommodation was responsible for myopia and that atropine causes temporary paralysis of the smooth ciliary muscle [8].

Anticholinergics are blockers of the action of acetylcholine at muscarinic receptors (MRs). Acetylcholine is a neurotansmitter that plays an important role in retinal development and regulates eye growth. The most effective treatment to slow the progression

of myopia is topical antimuscarinic medication, namely the use of low-dose atropine (0.01–0.05%) eye drops. However, the side effects of this medication include light sensitivity and blurring [2], so it is rarely prescribed [8].

The topical application of Pirenzepine does not block accommodation while decreasing the progression of childhood myopia [8,26].

### 3.3. Light in the Lives of Children and Adolescents

Outdoor illumination levels depend on weather, altitude, and latitude. The illuminance of a sunny day can be 150–130,000 lx, 50,000 lx on a hazy sunny day, and 15,000 lx on an overcast day. However, these levels are in stark contrast to typical indoor illumination levels, with values from around 1000 lx to values of 500–100 lx [27,33]. The effects of illumination on human refractive development occur within the context of the changing refractive state in the years after birth [33]. Circadian responses to light are thought to be mediated primarily by melanopsin-containing retinal ganglion cells, not rods or cones. Melanopsin cells are intrinsically blue-light-sensitive but also receive input from visual photoreceptors [34,35]. The scientific literature contains a large number of studies that relate circadian, neuroendocrine, and neurobehavioral responses to calibrated light exposures [35].

The light that enters through the human eye not only has the function of image formation but also influences the health and well-being of human beings by producing non-imaging effects in the long and short term (acute) [36]. From a light spectrum point of view, melanopsin in ipRGCs has a maximum of between 460 and 500 nm, while the visual system is more sensitive to mid-wavelengths of the visible spectrum at around 555 nm [37,38]. The duration, timing, spatial distribution, intensity, and power of the spectrally distributed light reaching the eyes can influence circadian rhythms and thus health [38,39].

Light does not affect children and adolescents in the same way as it does adults. For many days throughout the year, when children and adolescents come to school in the morning, they do so in total or partial darkness. The intensity of natural light entering classrooms through windows varies throughout the day and during different seasons. Students work for many hours in classrooms with lighting that does not resemble natural light in either composition or intensity. When they leave school, they usually spend very little time receiving natural light. This results in students living virtually every day with insufficient light, not receiving the proportion and amount of visible light for which we humans are genetically designed [40].

Teenagers usually go to sleep and get up several hours later than normal, and they have difficulty waking up in the morning for school. The reason could be the hormonal changes that occur at puberty. Schools do not have enough global lighting (artificial light and daylight) to stimulate their circadian system. This is more important in the months of less natural light. As teenagers spend more time indoors, they may lose the morning light necessary for circadian resetting. It would be advisable for the protection of the health of adolescents that they receive higher levels of morning light (or daylight) in schools and lower levels of night light (or daylight) at home [39].

School building spaces are the most important non-residential indoor environments, which often have, among other things, inadequate lighting [41]. The structural characteristics of building installations have a profound influence on learning. Inadequate conditions in classrooms are mentioned as factors relevant to poor student progress [4]. Ensuring good quality lighting in educational environments is complicated [42]. Student performance is better in classrooms with higher light intensity. Children can differentiate their light needs according to the task at hand [43]. In rooms where lighting is not uniform, with more luminosity over the immediate task area than the surrounding area, discomfort effects may be important [44]. There is extensive evidence of damage being caused to children's vision due to poorly lit classrooms [45].

A study was carried out with students who were fitted with a filter that did not allow the most important part of the light for the elimination of melatonin to pass through. Eleven students at a North Carolina school with unusually high levels of daylight were fitted with orange filter glasses to remove short-wavelength circadian light for five consecutive days. It was observed that the orange filter glasses allowed them to perform their activities; however, their dim light melatonin onset (DLMO) was delayed by half an hour compared to the previous week. Another study compared the behavior of students between two seasons, with significant differences in the amount of natural light received during the day. In this study of 16 students in New York, their DLMO was found to be delayed by 20 min and sleep output by 16 min in the spring compared to the winter [39].

### 3.4. Myopia and Light Insufficiency

Both the time spent outdoors and in sports are related to incident myopia. Researchers believe that outdoor time has the largest effect, independent of distance work activity [46] and physical activity levels [21].

In a comparison of myopia among children, it was concluded that the effects are related to educational pressure and time spent outdoors. Sports activities for hours in the summer months by both myopic and nonmyopic children can contribute to delay the excessive growth of the eye [47]. It has been observed that the progression of myopia is much faster in months with less daylight (winter) than in months with more daylight (summer) [48]. Seasonal effects (quantity of daylighting) are more powerful than more potent medicaments or optical treatments to try to reduce the progression of myopia [10].

Myopes have lower vitamin D levels than non-myopes (Donal Mutti et al., 2011). Studies to date have suggested that the relationships between factors that may lead to the onset of myopia, and vitamin D levels may be minimal [33].

A comparison of progression rates between Taiwanese school children in urban and rural areas showed that the average progression rate in urban areas was higher than in rural areas. Environmental factors such as urban development and academic level may be important factors contributing to myopic progression [13]. Se ha observado una disminución de la incidencia de la miopía en niños cuyas eddes estaban entre uno y tres años [9] después de aumentar el tiempo que estuvieron entre 1 y 2 horas diarias [49,50].

A decrease in the incidence of myopia has been observed in children whose ages were between one and three years [9] after increasing the time they were outdoors between 1 and 2 h per day [49,50].

Children living in countries like Singapore with a high prevalence of short-sightedness spend less time outdoors than children in countries like Australia with a lower prevalence of short-sightedness [27]. It has been observed that the prevalence of myopia in Chinese children aged 6–7 living in Sydney (3.3%) is significantly lower than in Singapore (29.1%). The reason for this difference in the prevalence of myopia is that children living in Sydney are much longer outdoors each week. Children in Taiwanese schools are required to perform at least 11 h of outdoor activities per week, and as a result, Taiwanese health authorities claim that shortsightedness has been reduced in all children. There is no need to receive high solar intensity for the prevention of myopia; however, longer outdoor activities with less intense sunlight intensity in corridors or under trees are preferable [12].

The approach of increasing time outdoors has been validated in school-based interventional research, with increases from 25 to 50%. The Myopia Control Programme in China specifies a period of one to two hours outdoors every day [9,50].

The spectral power distribution of the Sun is continuous, while that of the luminaires that we use are discontinuous. It has been found in experimental tests with animals that there are wavelengths of light that affect the refractive development of the eye and the growth of the axial length of the eye. Researchers say that when chicks receive intense light, whether it is daylighting or artificial light, there is a delay in the development of experimental myopia [17]. Exposure to violet light (VL, 360–400 nm wavelength) in particular, which is the shortest wavelength range of visible light, has a protective effect

against myopia by suppressing the elongation of the axial length (AL) [17,51]. As VL acts on the myopia-suppressive gene EGR1, it suppresses myopia progression and is therefore an important external environmental factor in controlling myopia. It induces a significantly higher up-regulation of EGR1 in chick chorioretinal tissues than blue light under the same conditions [51]. Violet light is abundantly present in sunlight, while it is rarely detected in indoor ambient light. If the VL content of natural light at different times of the day is compared with that entering through glass into indoor environments, the absence of VL behind the glass can be seen [17]. These results suggest that violet light is important for not only the prevention of myopia progression but also the onset of myopia [51]. Excessive ultraviolet (UV) protection and the absence of VL in artificial light may be possible reasons for the global myopia pandemic [17].

A systematic review of articles on this subject revealed great heterogeneity in the results, and it can be stated that increased outdoor time is effective both in preventing the onset of myopia and in slowing down the progression of myopic refractive error; however, paradoxically, outdoor time does not slow down the progression of myopia in eyes that are already myopic [52]. Most but not all prospective studies and cross-sectional surveys find an antimyopic effect with increasing outdoor time [26].

There have been research studies, although not statistically significant, in which red lighting illumination is provided to people during the afternoon (i.e., 15:00). People said they perceive a significant reduction in the feeling of drowsiness and a greater subjective sense of vitality. The use of luminaires that combine white LED with red light could give a sufficient stimulus to be able to work during the afternoon [53].

According to Mardaljevic's research, 24 h a day can be divided into three phases in relation to the dark–light cycle. The phase 6:00–10:00 a.m. is defined as circadian resetting, 10:00–18:00 is the period of alerting effects of daylight, and 18:00–6:00 the time of bright light avoidance and dim light only. The most important thing to get a circadian reset is to receive intense light during the morning. It is desirable that people are exposed to bright light during 10:00–18:00 for its potential to increase alertness [54,55]. It should be noted that there are important differences in the intensity in each part of the light spectrum of the day that we receive depending on altitude, latitude, and the season of the year. The color of the furniture is also a factor to be taken into account. The combination of interior lighting and daylight at each workplace will depend on the position of the worker in relation to the windows [41,56].

Although light reaching the retina may be the key, it is difficult to measure the W/cm$^2$ of each wavelength band entering through the pupil and reaching the retina [33]. Objective studies on the association between light exposure and myopia, not time spent outdoors, are needed to assess how much bright light is necessary for myopia prevention [57].

### 3.5. The Non-Visual Effects of Light on Humans

Light influences the hormonal secretion in our behavior and the regulation of circadian cycles [37]. The suprachiasmatic nucleus (SCN) is responsible for generating circadian rhythms that receive inputs from the intrinsically photosensitive retinal ganglion cells (ipRGCs) through the activation of the photopigment melanopsin. Dopamine (DA) is a neurotransmitter that has the ability to modulate the refractive development of the human eye depending on its concentration, which varies depending on the amount of light received, especially in blue range. DA has a circadian internal retinal clock regulating function [26].

Diurnal rhythms in ocular dimensions, rhythms in retinal signaling, and molecular biology are linked to refractive development. DA acts as an inhibitor of axial elongation [9]. Retinal dopamine (DA) messages affect many intraretinal processes of the light–dark cycle, including the entire light–dark adaptive state of the retina. The possible link of circadian rhythms in the control of ocular refraction could explain the effects of outdoor and light exposures on refractive development. Researchers believe that the retinal endogenous clock could have a connection with the light entering our visual system and the retinal Zeitgebers,

and therefore it could influence the development of the eyeball and the possibility of producing ametrophies if the appropriate amount of ambient light is not received at every moment of the day. This connection could give a biological explanation of the apparent positive action to curb the nearsightedness of daylight exposure [25].

Dopamine may play an important role in the refractive development of the human eye [58]. D2 (D2R and D4R) receptors are considered the most important myopia-related receptors within the two groups of dopamine receptors in the retina (D1 and D2) [17].

A significant number of publications state that exposure to sunlight and being outdoors stimulate the release of dopamine in the retina [17,27,58]. The release of dopamine is increased by receiving daylight or indoor lighting and decline without light [25,58]. Dopamine levels in the retina increase if we are exposed to intense or bright light. This could be an option to control the progress of myopia [11].

The dopamine metabolite dihydroxyphenylacetic acid (DOPAC) is generally accepted as an indicator of dopamine release. Midday retinal levels of this metabolite are 30% higher in our "elevated" light-level condition (15,000 lx·7.75 h per day) [33].

Chronodisruption is a common problem in modern societies because of our habits of life. The most developed societies have the highest prevalence of child myopia. The study of chronodisrupion could improve the knowledge of the mechanisms that produce myopia and open new fields of research and treatment to reduce the growth of the childhood myopia pandemic [25].

Since the discovery about 20 years ago of the non-visual methods of light absorption, it has been known that apart from the image-forming effects (IF) of light from which the criteria for correct lighting were developed, non-image-forming effects (NIF) of light exist. The discovery of NIF has enhanced researchers' belief in the importance of daylighting [59] and has raised new criteria to be taken into account for proper interior lighting [60]. Due to all of the factors mentioned above, the parameters to be met by a luminaire and its environment for the proper lighting of the workstation have to be modified and expanded [46].

*3.6. Regulatory Lighting Framework*

The technical characteristics of the luminaires and the lighting conditions at the workplaces where they are used are clearly defined in international standards [61,62]. The standards ISO 8995:2002 "Lighting in Workplaces" (2201/220) [61] and prEN 12464-1:2019 "Lighting in Workplaces" [62] are in the process of being reviewed. International regulations are beginning to take account of the NIF effects of light [62].

The recommended illuminances for classrooms are between 300, 500, and 750 lx, with recommendations for blackboards to be maintained at an illuminance of 500 lx in order to avoid specular reflections. Classrooms and tutorial rooms should be maintained at an illuminance of 300 lx.

Lighting requirements are determined by the satisfaction of three basic human needs: visual comfort, where workers have a sense of well-being, and indirectly also contributing to a high level of productivity; visual performance, where workers are able to perform their visual tasks, even in difficult circumstances and for longer periods; and safety [62].

The Illuminating Engineering Society of North America (IESNA) [63] has written indoor lighting proposals, in which the intensities vary depending on age, which is not found in the ISO and European standards.

Table 1 shows the recommendations for target illuminances (lx) based on the visual ages of the observers (years) for certain indoor education situations.

**Table 1.** Indoor lighting at workplaces. Lx per group of ages recommended.

| <25 Years Old | 25 to 65 Years Old | >65 Years Old |
|---|---|---|
| 200 lx | 400 lx | 800 lx |
| 250 lx | 500 lx | 1000 lx |
| 375 lx | 750 lx | 1500 lx |

The illumination must be controllable. Classrooms used for evening classes and adult education should be maintained at an illuminance of 500 lx [63].

The most important parameters to be taken into account for the manufacture and sale of luminaires can be taken from the CIE Technical Reports and the Commission's Delegate Regulation (EU) Nº 874/2012 from 12 July 2012 [64].

The photometric and chromatic tests outlined in EN 13032-4 [65] are some of the most important parameters to be taken into account, including the light emission or luminous flx emission, luminaire opening angle, correlated color temperature (CCT), color rendering index (CRI), and risk of blue light (according to EN 62471) [66]. Other factors are the energy consumption indicators and light source service life parameters. In a study carried out on LED luminaires purchased by consumer inspectors in commercial establishments in the Community of Castilla la Mancha in Spain, it was found that they meet the requirements set out by international standards. All of the luminaires analyzed were found to be in risk group 0 (no risk) [67,68].

### 3.7. New Indoor LED Lighting Luminaires

The differences between indoor and outdoor ambient lighting, such as the intensity and wavelength of modern electronic lighting equipment, may be a way to control myopia as an environmental factor [17].

Scientists around the world are proposing new luminaires, the spectral power distribution (SPD) of which is intended to resemble that of the sun in order to match our circadian cycles [69].

Several prototypes developed by researchers who claim that these different combinations of LED light improve performance are shown below and that they could serve to not alter our circadian rhythm.

Remote-controllable light source grouping, together with a quadruple-chip light-emitting diode driven by pulse width modulation currents to provide healthy lighting: RGB tricolor LEDs can emit pure white light. The color rendering index is of worse quality when the CCT is changed. The combination of white LED with the red–green–blue–white (RGBW). The authors of this research believe that the best solution for good interior lighting are RGBW LED (Figure 2) [70].

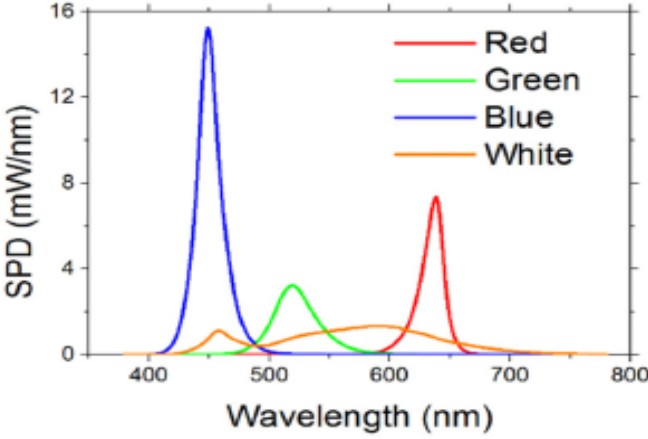

**Figure 2.** Parameters of the RGBW LED.

Other researchers think that the best option is the tunable LED lighting with five channels of RGCWW: red, green, cyan, warm white, and cool white (RGCWW). LEDs are individually controlled in this prototype. The researchers claim that, with this new source of indoor lighting, they managed to obtain dynamic daylight spectrum and complete mixing monochromatic thanks to the width and shape of the LED spectra, the green gap, and the computation capability. They can be very useful to illuminate interiors where not enough daylight comes in (Figure 3) [71].

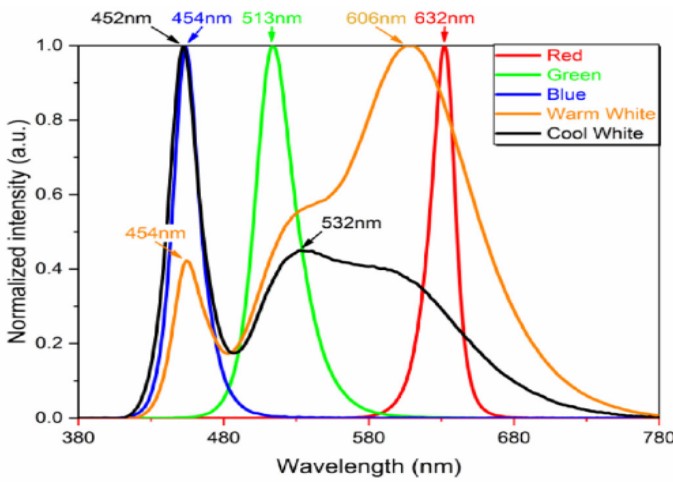

**Figure 3.** Tunable LED lighting with five channels of RGCWW.

Another approach involves spectral energy distribution with qualities close to daylight with a high CRI, with a continuous and balanced spectrum in the visible emission range, ideally harmonized with human sensitivity. The high blue energy part of the total blue range can be reduced to minimize the ups and downs of the spectral energy distribution. The spectral intensities in the visible range of three developed LED combinations are shown below (Figure 4) [72].

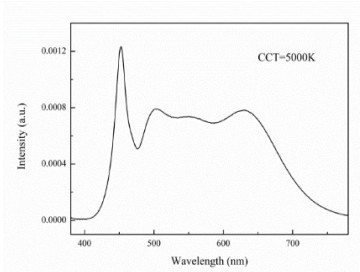

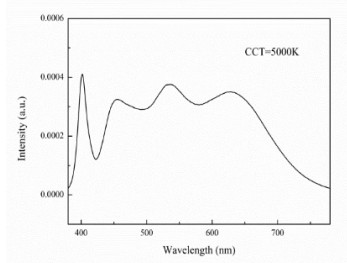

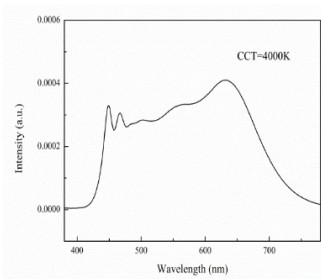

(**a**) The emission spectrum of 5000 K LED package under blue LED excitation.

(**b**) The emission spectrum of 5000 K LED package under violet LED excitation.

(**c**) The emission spectrum of 4000 K LED package under dual-wavelength blue LED excitation.

**Figure 4.** Intensity and spectral distribution of different combinations of LED visible light emitters [72].

The CIE, in its Technical Report CIE 015:2018, showed different spectral distributions. The next figure shows the relative SPD of each of the nine LED illuminants proposed in CIE Publication 015:2018, indicating their corresponding CCTs and distances to the Planckian locus in the UV space [73].

Figure 5 shows the SPD Standard D65 and D50, together with their corresponding indoor illuminants, ID65 and ID50. Also shown in the figure is the spectral transmittance of the average glass used to define the two indoor CIE illuminants [74].

Researchers have tried to reproduce daylight by creating light sources with five different types of light emitting diodes: red, green, blue LEDs; warm white light emitting LEDs and cold white light emitting LEDs (RGBWW) [71,75–77]. The circadian action factor (CAF), circadian stimulus (CS), or equivalent melanopic lx (EML) are some of the NIF parameters that researchers take into account for the design of new interior light sources [40,75,76,78].

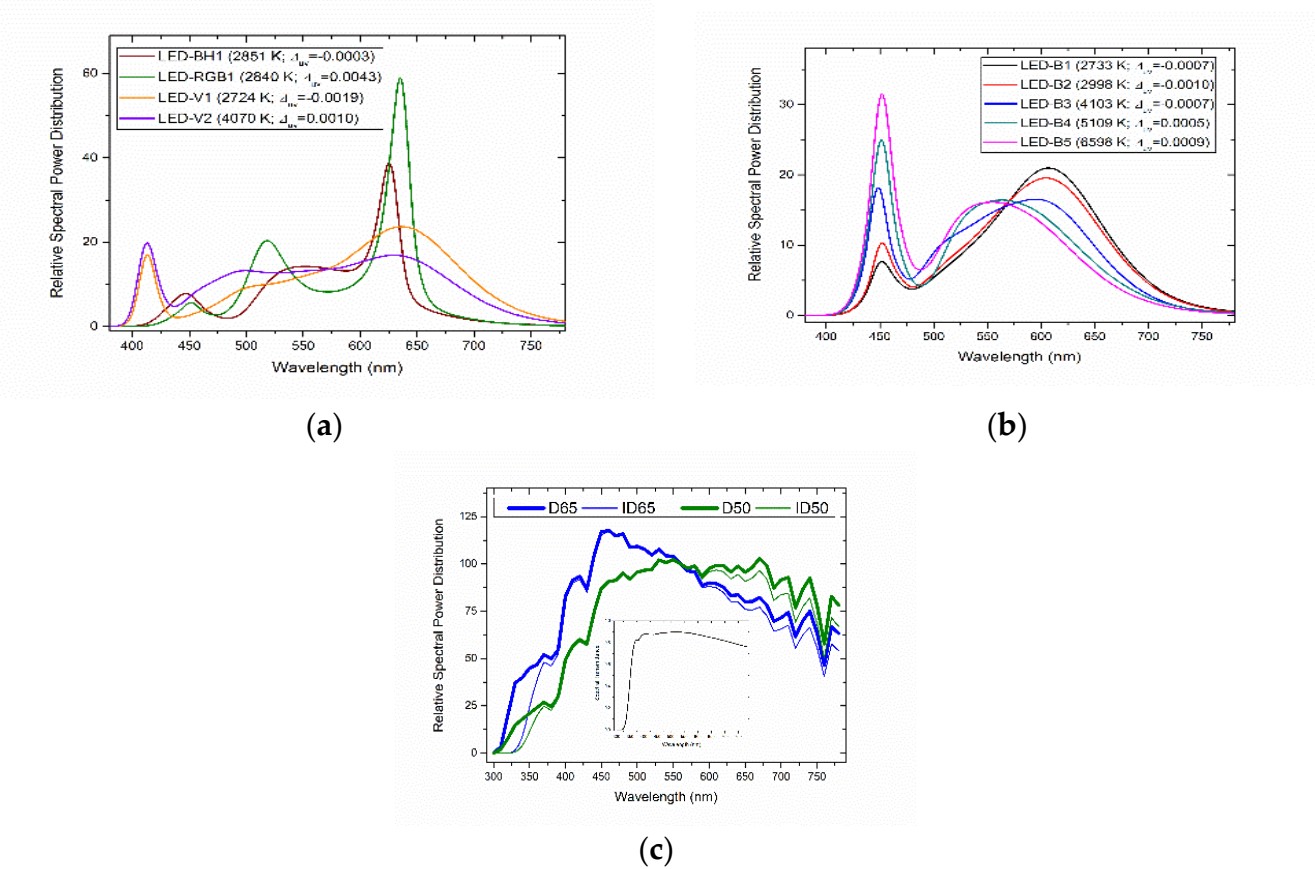

**Figure 5.** LED illuminants spectral power distribution of (**a**) the SPD, (**b**) the Standards D65 and (**c**) D50 [79].

If we compare daylight with new LED light sources, new indoor luminaries cannot produce an exact correlation with NIF effects of the light and the circadian cycle because of the differences between people and multiple environmental factors that cannot be controlled [68,77,79].

## 4. Discussion and Conclusions

This paper has shown the possible causes that the scientific world considers to be responsible for the current myopia pandemic. The authors of this article have found an extensive bibliography for each section presented in this review. It is not possible to mention them all, and the authors have tried to refer to the articles of the first quartiles.

There is increasing evidence implicating daylight and circadian cycles in the process of the emmetropization of the eye. The possible influence of light on myopia progression has been taken into consideration by the developed countries in Asia most affected by this pandemic [9].

These measures seem to us to be appropriate and novel in the face of this important child health problem; however, even so, we believe that they may not be sufficient due to the periods of the day when they are inevitably in enclosed spaces, both because they are in class or studying at home due to educational pressure and because of the use at close distances of electronic devices of all kinds in their leisure time. These activities are generally carried out in poorly lit environments, and therefore we believe that it is in this area that solutions should be sought.

Some authors claim that in addition to increasing outdoor time, the use of small doses of atropine, the use of rigid contact lenses to flatten corneal surfaces at night, and the use of prescription glasses and special contact lenses can reduce the progression of the myopia by 50.0% or more [2,9]. Our aim is to propose possible solutions that make their use and that of optometric correctors unnecessary.

For more than a decade, studies have proposed the importance of the blue part of light as a regulator of DA synthesis in the retina as a key factor in the appearance or otherwise of myopia [11,26,35,64], while others have claimed that it is found in the VL part of the spectrum [17,51]. The blue part centered at 490 nm is key in the regulation of circadian cycles; however, as outlined in recent CIE publications, the five photoreceptors present in the retina influence this regulation in a way that is still largely unknown [37].

Numerous studies by scientists studying IF and NIF effects of the light absorbed through our eyes have found an important role of each part of the visible light spectrum [33]. Scientific experts have recommended the amount and composition of light we are to receive in each part of the day [80]. Photoreceptors of the eye absorb light between the wavelengths 380 to 780 nm [81].

We propose that the education authorities consider that children should spend at least one hour a day in the open-air mornings performing sports or educational activities, especially during autumn and winter.

By simple evolutionary logic, the entire range of the electromagnetic spectrum to which our retinal photoreceptors are sensitive must have a positive function, whether known to researchers or not. In order to try to help slow down the current myopia pandemic, we believe that we should neither increase the intensity of indoor light sources nor increase the blue part of the spectrum in a massive and uncontrolled way, but rather try to replicate the spectral distribution and intensity of each part of the livable solar spectrum in the least imperfect way possible, and to analyze, in each interior space, the lack of intensity and spectral distribution, ranging from purple to the limit of red to infrared, from the natural light that can enter through windows (daylighting) and complement it with new combined LED light sources for interiors. It would be interesting to study how this can be replicated from 380 to 780 nm continuously with existing LED light sources, as well as to consider replacing window panes with materials that allow light to pass through from 380 nm. According to the progress of LED technology and studies of the emission of existing LED luminaires, we believe that it is possible to develop luminaires with a regulated irradiance so that their emission is higher in the blue range during the morning, in compliance with current international regulations.

The use of luminaires that combine white LED with red light could justify reducing the amount of blue light in the luminaires

On the basis of the research carried out for this study together with that of other publications currently under review, the research group of the degree in Architecture of the University San Pablo CEU is developing a utility model that will have an emission in the entire range of the visible in which the irradiance of blue and VL can be regulated for at least three different levels.

There are an enormous amount of variables that influence the light that enters through our eyes at every moment of the day, depending on such a large number of factors that it is not possible to manufacture a luminaire that can replace daylight [82], but if new LED indoor lighting is not developed with this range of visible light to see the biological responses, we will not be able to see if they can be a part of the solution for the pandemic of myopia.

**Author Contributions:** Conceptualization, D.B.M.; methodology, D.B.M.; formal analysis, D.B.M.; investigation, D.B.M.; resources, D.B.M.; data curation, D.B.M. and R.A.G.-L.; writing—original draft preparation, D.B.M. and R.A.G.-L.; writing—review and editing, D.B.M. and R.A.G.-L.; visualization, D.B.M. and R.A.G.-L.; supervision, R.A.G.-L.; project administration, D.B.M.; funding acquisition, D.B.M. and R.A.G.-L. All authors have read and agreed to the published version of the manuscript.

**Funding:** The authors wish to thank CEU San Pablo University Foundation for the funds dedicated to the Project Ref. USP CEU-CP20V12 provided by CEU San Pablo University.

**Conflicts of Interest:** The authors declare no conflict of interest.

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
