# Peer review of "Pandemic of Childhood Myopia. Could New Indoor LED Lighting Be Part of the Solution?"

_energies, doi:10.3390/en14133827_

Round 1
Reviewer 1 Report
General comments:
Dear authors,
Thank you for your submission.
- I question whether this work is a major novel addition to existing research on the topic? Below, are just two short abstracts from articles published in Nature Journal in 2018 and 2019, yet these were not even mentioned in your literature review.
“Alarming health consequences are emerging. For example, a global rise in short-sightedness (myopia) since the 1960s has been linked to low exposure to daylight . Today, around 70–80% of young adults living in Taiwan, Japan, Hong Kong, Singapore and other parts of eastern Asia are short-sighted; by contrast, in 1950s China, only 10–20% of the population was affected. By 2050, half of the world’s population could be myopic. Yet myopia might be prevented by spending just 2 hours a day outdoors in bright sunlight. Researchers are trying to pin down the particular biological mechanisms involved.” doi: https://doi.org/10.1038/d41586-019-01238-y
“Independent research — beyond the lighting industry — is needed into the health and environmental impacts of LED sources, including those with adjustable spectral characteristics, intensity, timing and duration based on the time of the day, evening or night. Emissions outside the visible range must be considered, such as near-infrared radiation (750–950 nm) that is present in daylight and incandescent lamps but not LEDs. Research shows that there needs to be a balance - the use of these light frequencies can repair damaged retinal cells and are necessary.“ doi: https://doi.org/10.1038/d41586-018-00568-7
Also see this research:
https://www.preprints.org/manuscript/202012.0037/v1
https://doi.org/10.3390/clockssleep2010008
- What is your research question or hypothesis?
- I am missing the “Result” section in your manuscript.
According to the MDPI template: “This section may be divided by subheadings. It should provide a concise and precise description of the experimental results, the interpretation, as well as the experimental conclusions that can be drawn.”
- I am missing the “Materials and Methods” section in your manuscript. According to the MDPI template: “The Materials and Methods should be described with sufficient details to allow others to replicate and build on the published results.”
Detailed comments
L1
Article type: This manuscript reads more like an essay, or a commentary, or a book chapter, not as an MDPI’s article. As this is supposed to be a review, it needs to read like one.
L2-3
Your title indicates that you will answer in detail the question connected to new indoor LED lighting as a part of a solution for a pandemic of childhood myopia. Unfortunately, this was not covered in your paper. In fact, this important part is missing.
L11-23
It’s important to MDPI’s abstract template structure which includes this information:
“(1) Background: Place the question addressed in a broad context and highlight the purpose of the study; (2) Methods: briefly describe the main methods or treatments applied; (3) Results: summarize the article's main findings; (4) Conclusions: indicate the main conclusions or interpretations. The abstract should be an objective representation of the article and it must not contain results that are not presented and substantiated in the main text and should not exaggerate the main conclusions.”
L21
It’s necessary to provide a proper literature review, based on PRISMA methodology for a review paper: http://prisma-statement.org/ However, this is missing. You even didn’t mention the number of articles you analysed and the search keywords etc. and which databased were searched etc.
L20-21
This statement is false. In order for artificial lighting to mimic the natural light wavelengths outside the visible spectrum should be introduced - and here is the problem. From a technological point of view, it is not possible to introduce NIR and IR into LEDs, as the electronics will overheat.
L216
I think you mean “continuous”??
L221-223
What about macular degeneration and blue light hazard?
See: https://pubmed.ncbi.nlm.nih.gov/16445433/
L231-232
This is due to a lack of IR light.
L302
Please correct this typo. Also, it would be better if this section is called “Regulatory lighting framework”, and divided into subsections. A table would be useful, synthesising various characteristics.
L304-306. If one refers to a lighting standard, one should always check its correct English title and if the standard is current or in a draft form, as it is in this case. Also, if you refer to the IES standard, it should be quoted with the whole name and a link.
L386-390
Please correct your language style to be more scientific.
L429-430
This part of text should be removed.
L436
Please correct the heading here. Also, you should use the correct MDPI reference style.
Author Response
Red color: responses to revisor 1

Reviewer 2 Report
The article covers an interesting topic.
There are a few issues to consider:
- line 106: "T The", please revise (the issue occurs also in other part of the paper);
- line 191: ". 50]", please revise. (the issue occurs also in other part of the paper);
- line 253: please revise the unit of measure;
- line 302: please revise the title of the paragraph;
- line 317: why you not reported a caption? What do all the rows refer to?;
- line 360: please revise the resolution of the figures. Please add the number of the figure in the caption;
- figure 4 and 5, please revise the resolution.
Author Response
Blue color: responses to revisor 2
Round 2
Reviewer 1 Report
Dear Authors,
Thank you for your feedback and incorporating some of the comments provided from the first review. My major concern still involves Chapter 2. Materials and Methods for this literature review.
I appreciate the fact that you added this new chapter and included some background information on the process you followed for your literature review, however, this is not fully consistent with PRISMA guidelines.
Please refer to the following checklist at this link: http://prisma-statement.org/documents/PRISMA_2020_checklist.pdf
As an example of what is required, you can look at this paper: https://www.mdpi.com/2071-1050/12/15/5900/htm Chapter 2 and following table:
Table A1. Codification of Papers included in the Systematic Review
I am also missing a PRISMA flow diagram which should be part of the text.
You can use this generator: http://prisma.thetacollaborative.ca/
At the moment, your information would not allow for repeatability of this study, therefore, until it’s improved, I cannot accept it.
Also, please check the information below, which explains why Google Scholar should not be included in the search:
https://onlinelibrary.wiley.com/doi/full/10.1002/jrsm.1378
Kindly remove it.
Detailed comments:
Line: 377-378: “It is not possible to mention all relevant articles written on this topic. We apologize if readers feel there is one missing that they consider important..” Please correct your language style to be more scientific.
Line: 287: The discovery of NIF has enhanced researchers' belief in the importance of daylighting [x] and has raised new criteria to be taken into account for proper interior lighting [xx].
[x] Please add this reference here: https://doi.org/10.1038/d41586-019-01238-y
[xx] Please add this reference here: https://doi.org/10.1038/d41586-018-00568-7
Author Response
We would like to thank again the Reviewer 1 for their detailed comments and suggestions for our manuscript. We firmly believe that their comments have been very useful in order to identify important areas which required improvement. After completion of the edition, the revised manuscript has benefited from such an improvement in its overall presentation and clarity. Please find below a point-by-point description of how each comment is addressed within the manuscript. The original reviewer’s comments are in black and our responses are in red

Round 3
Reviewer 1 Report
Extensive editing of English language and style required!